# The Role of NLRP3 Inflammasome Activation and Oxidative Stress in Varicocele-Mediated Male Hypofertility

**DOI:** 10.3390/ijms23095233

**Published:** 2022-05-07

**Authors:** Giulia Poli, Consuelo Fabi, Chiara Sugoni, Marina Maria Bellet, Claudio Costantini, Giovanni Luca, Stefano Brancorsini

**Affiliations:** 1Department of Medicine and Surgery, University of Perugia, 06132 Perugia, Italy; chiara.sugoni@alice.it (C.S.); marinamaria.bellet@unipg.it (M.M.B.); costacla76@gmail.com (C.C.); giovanni.luca@unipg.it (G.L.); stefano.brancorsini@unipg.it (S.B.); 2Department of Surgical and Biomedical Sciences, Urology and Andrology Clinic, University of Perugia, 05100 Terni, Italy; 3Department of Medicine and Surgery, University of Perugia, Via Mazzieri 3, 05100 Terni, Italy; consuelofabi93@gmail.com

**Keywords:** NLRP3 inflammasome, varicocele, male hypofertility, reactive oxygen species, antioxidants

## Abstract

Varicocele (VC) is the most common abnormality identified in men evaluated for hypofertility. Increased levels of reactive oxygen species (ROS) and reduced antioxidants concentrations are key contributors in varicocele-mediated hypofertility. Moreover, inflammation and alterations in testicular immunity negatively impact male fertility. In particular, NLRP3 inflammasome activation was hypothesized to lead to seminal inflammation, in which the levels of specific cytokines, such as IL-1β and IL-18, are overexpressed. In this review, we described the role played by oxidative stress (OS), inflammation, and NLRP3 inflammasome activation in VC disease. The consequences of ROS overproduction in testis, including inflammation, lipid peroxidation, mitochondrial dysfunction, chromatin damage, and sperm DNA fragmentation, leading to abnormal testicular function and failed spermatogenesis, were highlighted. Finally, we described some therapeutic antioxidant strategies, with recognized beneficial effects in counteracting OS and inflammation in testes, as possible therapeutic drugs against varicocele-mediated hypofertility.

## 1. Introduction

Varicocele (VC) is an anormal dilation of the scrotal venous pampiniform plexus that drains blood from each testicle. VC occurs in around 20% of males, and 40% of infertile males. In at least 85% of cases, VC is left-sided, while right-sided cases are rare [1], and is the most frequent cause of a low sperm count, abnormal semen analysis, atypical sperm morphology, and decreased sperm motility [2]. Although there is no doubt about the clinical association between VC and male hypofertility, it is not clear how VC impairs the production, structure, and function of sperm [3]. At the clinical level, VC is surgically correctable, but not all men have positive results, in terms of fertility potential, after varicocelectomy [4].

It is generally accepted that the pathogenesis of VC is complex and multifactorial. The synergy of genetic and other factors, such as scrotal hyperthermia, testicular microcirculation disturbance, hypoxia, oxidative stress (OS), and nutrient deprivation, occurs during VC disease [5].

In particular, reactive oxygen species (ROS) are generated by biological systems during OS. Examples of ROS include superoxide radicals (O_2_^•−^), hydrogen peroxide (H_2_O_2_), hydroxyl radicals (•OH), and singlet oxygen (^1^O_2_) [6,7]. Several processes, such as protein phosphorylation, the activation of several transcription factors, apoptosis, immunity, and differentiation, increase ROS production, which then react with important cellular structures such as proteins, lipids, and nucleic acids [8]. A large body of evidence shows that OS stimulates the progression of several diseases such as cancer, diabetes, metabolic disorders, atherosclerosis, and male infertility [9]. It is reported that OS is one of the most important causes of male infertility, due to its adverse effects on both the structural and functional integrity of the sperm, leading to a failed spermatogenesis [10].

OS leads to a homeostatic imbalance between antioxidant capacity and ROS production in seminal plasma. Basal ROS concentration is essential for physiological sperm maturation, capacitation, acrosome reaction, and fertilization, but ROS overproduction was observed in men with VC disease. Consequently, sperm parameters, male fertility, and pregnancy outcome are strongly compromised in these patients; in addition, lipid peroxidation, DNA fragmentation, and apoptosis can ensue when the OS condition takes place, thus, causing damage to reproductive cells and decreasing semen quality [11,12]. Moreover, OS plays a crucial role in the development and perpetuation of inflammation [13].

In particular, several studies highlight the prominent role of ROS in NLR pyrin domain containing 3 (NLRP3) inflammasome activation, the best characterized oligomeric molecular complex that activates innate immune responses by generating proinflammatory cytokines, including IL-1β and IL-18 [14]. Recently, a two-signal model for the activation of NLRP3 was described: the priming signal (signal 1) and the activation signal (signal 2) [15].

The priming signal is activated by damage-associated molecular patterns (DAMPs), which come from endogenous cells under stress, and by pathogen-associated molecular patterns (PAMPs), which are specific to microbial invaders. In both cases, the binding between molecules and receptors results in the nuclear translocation of nuclear factor kappa-light-chain-enhancer of activated B cells (NF-kB), and consequently, the up-regulation of NLRP3 and pro IL-1β [16,17].

The priming signal is followed by the activation signal, which is due to alterations in ionic flux, mitochondrial DNA (mtDNA) release, ROS production, and lysosomal damage and dysfunction [18].

Although the exact mechanism of NLRP3 activation induced by ROS remains controversial, a model of ROS-dependent inflammasome activation was proposed, involving the thioredoxin-interacting protein (TXNIP). TXNIP is a thioredoxin (TRX) inhibitory protein, an important redox molecule, which behaves as a reductor of thiol and intracellular ROS level regulator [19]. The inhibition of TRX implies an inflammatory state, with lethal consequences for cells [19]. TXNIP is particularly expressed in the testis and human Sertoli cells (SCs), and is involved in a number of diseases, including infertility [20]. Under normal conditions, TXNIP is associated with TRX protein, and forms a multiprotein complex. The TXNIP–TRX detachment occurs following an intracellular ROS increase, and the disengaged TXNIP binds NLRP3 to its leucine-rich repeat domain, which determines the inflammasome activation, and results in the clavation and activation of caspase-1, IL-1β, and IL-18 [19].

In VC, the production of ROS, and the activation of the NLRP3 inflammasome pathway, triggers several pro-inflammatory cytokines [21], generating an inflammatory state with consequent damage to the male reproductive tissue, and the recruitment of immune cells producing reactive species, which further exacerbates OS (Figure 1).

In the first half of this review, we explored how abnormal testicular function and failed spermatogenesis are caused by ROS-driven NLRP3 activation and the overexpression of cytokines. In the second half, we highlighted alternative perspectives, by providing ideas for future research and potential treatment of VC.

## 2. OS and Inflammasome Activation in the Pathogenesis of VC

### 2.1. Testis Immunity and Inflammatory Response

The testicular immune microenvironment is extremely reactogenic. Indeed, mammalian testis present a unique immune setting that is essential for the maintenance of all testicular functions [22]. The physiological homeostasis of the testicular immune microenvironment is supported by Leydig and SCs [23,24]. SCs in particular, by virtue of a tight junction formation, constitute the main component of the blood testis barrier (BTB), and control the testicular homeostasis compartment by modulating the production of several growth factors and the development of prospermatogonia within the testicular niche [25]. In an androgen receptor-deficient murine SCs model, the barrier that prevents contact between the systemic immune cells and spermatogenic cells in the testicular stroma is destroyed [26]. As a result, mice develop an abnormal permeability to different BTB proteins, such as claudin-11, and a testicular homeostasis imbalance.

In the testis, pattern-recognition receptors (PRRs) are expressed by both Leydig and SCs, and play a key role in the innate immunity needed to counteract bacterial infections. At the same time, the presence of immunosuppressive testicular cells contributes to the secretion of regulatory factors that block the immunologic response [27]. Several studies [28,29] show that testicular macrophages (TMs) are the major immune cells producing cytokines that modulate testosterone synthesis [30] and spermatogenesis [31].

Recent studies, based on transcript and protein expression analysis, suggest that NLRP3 has a testicular role. In particular, in the testis of rodents and primates, the major site of NLRP3 expression are the SCs, and alterations in NLRP3 expression might impair fertility [32]. NLRP3 is also found in the peritubular cell, a further somatic cell type of the human testis, whose main function is testicular immune surveillance, and which can, therefore, contribute to the development of the sterile inflammation associated with male infertility. Thus, the presence of NLRP3 could be linked to the function of these cells [33].

### 2.2. OS-Induced Inflammation in VC

VC causes spermatic vein ischemia, which increases the production of nitric oxide (NO) and inflammatory cytokines [34]. TMs are polarized into M1, secrete high levels of IL-1β and TNF-α [35], and promote the NLRP3 inflammasome assembly complex, leading to the inhibition of normal testicular functions, such as spermatogenesis and testosterone synthesis [13,36,37]. While a moderate amount of interleukin IL-1β maintains the physiological function of testicular cells, such as SCs and Leydig cells, [38], the over-expression of other cytokines, including IL-37 and IL-18, in seminal plasma from VC patients is aroused [39]. Moreover, a significantly elevated level of NLRP3 protein in seminal plasma in VC patients, versus control subjects, is reported [13].

Inflammatory response activation induces a leucocyte recruitment that, in turn, enhances the ROS and OS levels by up to 1000 times, compared to normal conditions. Therefore, OS stimulates NLRP3 and the production of cytokines, that, in turn, further increases ROS production, up to and exceeding the availability of antioxidant systems, causing spermatogenic failure and the impairment of testicular function [40]. In detail, ROS have a negative effect on sperm by inducing lipid peroxidation, sperm DNA fragmentation, mitochondrial dysfunction, and apoptosis.

#### 2.2.1. Lipid Peroxidation

During redox reaction with an alternating carbon–carbon double bond, polyunsaturated fatty acids (PUFAs), which are particularly present in the sperm cell membrane, transform into lipid radicals. The elevated amount of PUFAs in sperm cells is the reason for their high sensitivity to OS, in respect to other cells [41]. Moreover, the large surface area with a reduced amount of cytoplasm also makes sperm cells more susceptible to ROS [42]. Indeed, a recent study comparing sperm parameters with levels of NO and antioxidant indices, such as ascorbic acid (AA) and glutathione (GSH), between infertile men and control groups, reveals sperm quality decreases in the infertile male group, whereas NO, GSH, and AA levels significantly increase, possibly to counteract the negative effect of ROS, and limit lipid peroxidation [43].

Excessive lipid peroxidation also causes mitochondrial membrane damage and dysfunction [41,44], resulting in the activation of caspases and, eventually, apoptosis. During apoptosis, cytochrome c is released, further increasing ROS levels, DNA fragmentation, and damage [45,46]. Furthermore, considering that mitochondrial oxidative phosphorylation occurs in the inner mitochondrial membrane, and provides ATP production for the flagellar movement of spermatozoa, its dysfunction reduces ATP production, with a decreased motility of male gametes [47]. In conclusion, the damage induced by PUFAs’ double bonds reduces the fluidity of the sperm membrane and impairs capacitation, acrosome reaction, and sperm–oocyte fusion [41,44].

The relation between NLRP3 and lipid peroxidation was evaluated in an in vivo rodent model of abnormal semen quality following a spinal cord injury (SCI). The results show that the consequences of a SCI include improved ROS production, lipid peroxidation, and inflammasome activation, which are directly correlated to decreased semen quality [48,49]. In this model, the concentration of NLRP3 and of malondialdehyde, a marker of lipid peroxidation, are correlated and found directly proportional to each other. Future studies will help understand the role played by lipid peroxidation and inflammasome activation in VC patients.

#### 2.2.2. Mitochondrial Dysfunction

Mitochondrial proteins involved in the electron transport chain (ETC) have a significant role in sperm quality [50,51]. Remarkably, 22 mitochondrial proteins identified in a total of 141 proteins, were differentially expressed in patients with VC, compared to control patients. Defects in sperm mitochondrial ultrastructure, induced by excessive endogenous ROS production, are associated with reduced ATP synthesis, sperm dysfunction, and decreased sperm mobility [52,53].

Recent studies reveal the direct link between mitochondria and NLRP3 inflammasome activation. NLRP3 activators induce mitochondrial destabilization with externalization, or the release of mitochondria molecules that directly bind and activate NLRP3, such as cardiolipin and mtDNA. NLRP3 then translocates in proximity to the mitochondria, where it contributes to the inflammasome platform assembly [54,55].

Additionally, ATP, a NLRP3 secondary activator, released during mitochondrial dysfunction and apoptosis, also promotes the release of oxidized mtDNA into the cytosol, which in turn interacts and activates the NLRP3 inflammasome [56]. Specifically, spermatic mtDNA is easily harmed by ROS, due to its circular structure with few DNA base pairs, shortage histones, and the absence of nucleotide excision repair (NER) [57]. Deletions in spermatozoa mtDNA are reported in men with VC [58,59,60]. A study, conducted by Gashti, shows that 81.7% of patients with VC have a deletion of 4977 base pairs in the mtDNA of sperm, while only 15.5% of the controls have the same deletion. In comparison with the nuclear DNA, mtDNA is highly sensitive to the oxidative damage associated with the VC phenotype [61]. In particular, the mtDNA regions most affected by deletions are those with genes encoding enzymes of the electron transport system, causing defects in ATP production, sperm immobilization, and infertility [62].

#### 2.2.3. Chromatin Damage, Sperm DNA Fragmentation and Apoptosis

Due to the high susceptibility of sperm DNA to OS, its alteration is a significant factor contributing to ROS-induced infertility [42,63]. It is well known that DNA is damaged by ROS, by either a direct or indirect process, through the mutagenic products of lipid peroxidation, such as MDA [64,65]. ROS cause different types of direct DNA damaging, including a single or double-strand break, DNA fragmentation, the creation of free base sites, changes in nitrogenous bases, and DNA cross-linking [66]. The consequence of these changes is the alteration of fundamental cellular processes, including DNA replication, transcription, transduction, and genomic instability [67]. Therefore, patients with severe DNA oxidative damage have deteriorated sperms, incomplete sperm maturation, sperm DNA fragmentation (SDF), and an increased apoptotic rate in germ cells and spermatozoa [42,68,69].

In particular, a close relationship between ROS and SDF is well documented [57]. Conversely, a direct link between NLRP3 and SDF has not yet been investigated. Verma and colleagues explore the connection between SDF and inflammation in VC, and analyze the concentration of many inflammatory mediators. They demonstrate that an increased expression of IL-10 limits the OS condition, through the inhibition of TNF-α, and IL-1α/β e IL-6 [70].

The increased apoptosis of germ cells and spermatozoa in VC can be associated with both the impairment of spermatogenesis mechanisms, and increased ROS production in the mitochondria [71,72,73]. The final step of the activation of caspase-3 and caspase-8 leads to protein cross-linking, cytoskeletal and nuclear degradation, SDF, and intracellular apoptotic body formation [74]. Recent evidence shows that in sperm cells, the apoptotic process is negatively regulated by phosphatidylinositol 3-kinase, an intrinsic pathway inhibitor. Indeed, the inhibition of this pathway induces the activation of the apoptotic cascade in sperm cells, with the consequential rapid motility loss, mitochondrial ROS generation, caspase activation in the cytosol, cytoplasmic vacuolization, and oxidative DNA damage [75].

## 3. Possible Therapeutic Approaches for VC Targeting OS and Inflammation

As previously mentioned, any alteration in the testicular immune microenvironment determining ROS increase plays a key role in male infertility pathophysiology [76]. Different studies identify a correlation between reduced antioxidant and anti-inflammatory activity, and excessive ROS production in the testis of VC patients [77]. Indeed, Agarwal and colleagues identify a significantly lower total of antioxidant capacity levels in the semen of infertile VC patients, associated with elevated ROS levels, in a meta-analysis study [78]. Antioxidants reduce OS and DNA damage in men, treat VC-associated human male infertility, and protect seminal plasma from oxidative damage and premature sperm maturation [79]. Therefore, to maintain the optimal functioning of sperm cells, it is necessary to balance the redox potential through antioxidant and anti-inflammatory systems, and it is crucial they can cope with an excess in ROS production [80]. In physiological conditions, testes present antioxidant mechanisms that neutralize free radicals, and may protect spermatozoa against damage caused by OS. These include not only the main ROS processing enzymes, but also small molecular weight antioxidant factors, including ions and free radicals scavengers [81]. A perturbation of just one of these antioxidant systems in pathological conditions leads to a limited antioxidant capacity, and induces a state of OS in the testes. Thus, an oral antioxidant therapy could restore the imbalance of free radical production, improving spermatogenesis vital elements and seminal plasma clearance capacity [80]. Indeed, a variety of antioxidants were assessed for their ability to counteract OS in the testes. In particular, numerous studies use different types of antioxidants as mono- or poly-formulations, included pharmacologically active herbal extracts, as therapy for men with hypofertility linked to VC [80]. Among the most effective antioxidant treatments are non-enzymatic factors such as resveratrol (3,5,4’-trihydroxystilbene, RSV), vitamins (vitamins E and C), coenzyme Q10 (CoQ10), and lycopene [80], and therapeutic drugs, including kallikrein, pentoxifylline, and cinnoxicam. All mentioned antioxidants are summarized in Table 1.

### 3.1. Resveratrol

As mentioned earlier, nowadays, the use of antioxidant natural compounds as possible substances that prevent or ameliorate male infertility arouses great interest [80]. Among these, RSV is one of the most investigated natural polyphenolic compounds. RSV, a polyphenolic phytoalexin, is produced by grapes and a few other plants, such as mulberries [82]. Several reports reveal that RSV displays a range of pharmacological properties, including anti-inflammatory, anti-oxidative, anti-thrombotic, and anti-proliferative qualities [83]. Hajipour and coworkers investigate the important antioxidant role of RSV in decreasing the testis tissue damage, in an animal model study. In particular, they test its protective effect on NLRP3 complex activation and apoptosis in an experimental VC rat model [82]. The results show a significant up-regulation of NLRP3 gene expression, after 3 months of VC induction. On the contrary, the NLRP3, ASC, and caspase-1 mRNA levels are significantly lower in VC rats supplemented with 20 mg/kg and 50 mg/kg RSV, when compared to non-treated animals. Moreover, they also found that the RSV-supplemented rat groups display reduced apoptotic markers compared to the control group. These data strongly suggest that RSV ameliorates VC-induced inflammation, and prevents apoptosis after VC in a dose-dependent manner [82]. Although many studies investigate the beneficial effects of RSV on animal models, there are still few human clinical trials that study the impact of this compound on male fertility. For example, a recent work evaluates the effects of GENANTE®, a multivitamin supplement containing 150 mg of RSV, in 20 patients with idiopathic infertility before and after 1, 3, and 6 months of treatment. A statistically significant improvement in total sperm count, sperm concentration, total motility, and progressive motility is reported after 1, 3, and 6 months of treatment, confirming the protective potential of RSV on sperm quality [84].

### 3.2. Vitamins

Recent literature highlights the importance of vitamins as powerful biological antioxidants, with positive effects on the semen of infertile VC patients [85]. Busetto and colleagues investigate the efficacy of vitamin supplementation as the primary or adjuvant treatment in infertile men with clinically diagnosed VC [86]. In this work, 45 patients with VC Grade I–III and oligoasthenoteratozoospermia (OAT) are enrolled and subsequently randomized in placebo *(n* = 24) and multivitamin-supplemented (*n* = 21) groups. Spermiograms are repeated after 6 months, and compared between the two experimental groups. The results show a significant improvement in total sperm count and progressive motility in the vitamin supplementation therapy group [86]. Several types of vitamins are known for their positive effects on testicle and sperm quality, and integrative therapy with these natural compounds improves sperm function, reducing OS damage. For example, vitamin E (VitE) is a fat-soluble vitamin that protects the cellular membrane and neutralizes radical species by forming stable non-radical products [87]. In seminal plasma, VitE acts as the primary antioxidant component of the spermatozoa, preventing DNA damage caused by ROS. Ener and colleagues investigate the antioxidant impact of VitE in the semen of a cohort of 45 infertile patients with VC and who underwent subinguinal varicocelectomy [88]. Patients are randomized into a group (*n* = 22) receiving VitE supplementation for 12 months, and a control group (*n* = 23). The pre-operative and post-operative sperm count and motility are compared, and results show an increment in these specific semen parameters in the group undergoing VitE supplementation [88]. Another study investigates the specific role of vitamin C (VitC) in VC patients after surgery [89]. A total of 115 patients with abnormal semen analysis are recruited. After surgery, patients are divided into two groups: a group (*n* = 46) administered with VitC supplementation for three months, and a control group (*n* = 69) using placebo therapy. The seminal parameters (sperm count, motility, and morphology) are analyzed before and after surgery, and compared between the two experimental groups. The post-operatory analysis shows a 30.4%, 19.1%, and 12.2% worsening of the three parameters, respectively. Analysis of seminal parameters in the group with VitC supplementation shows a restoration in sperm motility and morphology, compared to the placebo group [90]. These vitamins, acting as free radical scavengers, also contribute to immune defense, by modulating inflammatory genes and the inflammasome complexes. Indeed, in a 2016 study, researchers show the inhibitory potential of VitC on NLRP3 inflammasome and IL-1β activation through scavenging of mitochondrial ROS, in vitro, and in a septic shock murine model [89]. Mouse bone marrow-derived macrophages (BMDMs) are treated with different concentrations of lipopolysaccharide (LPS) + ATP to induce NLRP3 inflammasome activation. Different concentrations of VitC are tested on the BMDMs and the inhibitory effect on NLRP3 inflammasome activation analyzed. Result show a decrease in IL-1β secretion in a dose-dependent manner, indicating that the NLRP3 inflammasome activation is inhibited by VitC [89].

### 3.3. Coenzyme Q10

CoQ10, also known as ubiquinone, is a fat-soluble, antioxidant, isoprenylated benzo-quinone, synthesized from tyrosine, which prevents oxidation and lipid peroxidation [91]. It is also involved in the regulation of mitochondrial redox reactions, and the electron transport in the mitochondrial respiratory chain for the synthesis of ATP [91]. In cells, CoQ10 presents three redox states, depending on its ability to accept or donate electrons: ubiquinol (CoQ10-H2-reduced form), ubiquinone (oxidized form), and semiquinone (a radical) [92]. CoQ10 is notably biosynthesized in the testes; in fact ubiquinol is particularly present in sperm, and its concentration in seminal fluid shows a correlation with semen parameters [93]. Several clinical studies report the beneficial effects of CoQ10 supplementation on the sperm parameters of VC patients [94]. For example, in a pilot clinical trial, Festa investigate the effect of an oral CoQ10 supplementation therapy, analyzing sperm parameters and seminal plasma total antioxidant capacity (TAC) of male VC patients. In this study, 38 patients are enrolled, and the diagnosis of VC clinically confirmed by color Doppler ultrasonography [95]. The results show an increase in the semen parameters after CoQ10 supplementation. Sperm forward motility, ejaculate volume, and sperm density increases about 8%, whereas seminal plasma TAC shows a mean increase of 40% [95]. Despite the lack of literature about the modulatory potential of CoQ10 on NLRP3 in VC patients, several studies confirm its anti-inflammatory potential in other pathologies. In a 2018 study, Chokchaiwong and colleagues investigate the ability of CoQ10 to restore mitochondrial function, and to prevent oxidative damages and NLRP3 activation in multiple acyl-CoA dehydrogenase deficiency (MADD), an autosomal recessive disorder that causes mitochondrial dysfunction [96]. They treat lymphoblastoid cells of MADD patients with different concentrations of CoQ10 and data shows a significant restoration in mitochondrial function and ROS, lipid peroxidation and a decrease in the level of IL-1β. These results confirm the regulatory activity of CoQ10 on NLRP3 activation [96].

Regarding the reduced form of CoQ10, several studies demonstrate its beneficial effects as a potent lipophilic antioxidant in mitochondrial and lipid membranes [97]. Moreover, as reported by many works, one of the most prominent advances in the pharmaceutical formulation of CoQ10, in its reduced form, is increased bioavailability, up to 4.8-fold greater in comparison with its oxidant form [98,99,100,101].

An additional advantage of ubiquinol supplementation is bona fide antioxidant activity, while the oxidant version of CoQ10 requires enzymatic reduction before acting as an antioxidant [102]. Alahmar and colleagues demonstrate that the oral administration of ubiquinol to males with idiopathic oligoasthenoteratozoospermia improves semen quality, and the oxidative status in seminal plasma [103].

In a double-blind study, Safarinejad and co-workers analyze the effect of ubiquinol on the semen quality of infertile men. A total of 228 patients are randomized into a treated group (*n* = 114), and receive a daily oral ubiquinol therapy (200 mg) for 26 weeks; a placebo group (*n* = 114) receive a similar regimen. At the end of the treatment phase, analysis of seminal parameters, such as sperm density, sperm motility, and sperm morphology, are performed after another 12 week off-drug period. The results show a significant effective improvement in sperm density, sperm motility, and sperm morphology in patients treated with supplemented ubiquinol, suggesting its efficacy in the treatment of infertile man [97].

### 3.4. Lycopene

Lycopene is a naturally synthesized, red-pigmented, unsaturated linear carotenoid, particularly present in fruits and vegetables such as tomatoes, papayas, watermelons, apricots, pink grapefruits, and rosehips [104]. This natural compound is noted as a major redox human molecule, present at elevated concentrations in seminal plasma and the testis of healthy males, 10 times higher than in other tissues [105,106]. Recently, the importance of lycopene is highlighted as a therapeutic strategy against male infertility. In an experimental study, Babaei investigates the antioxidative effects of lycopene on an experimental VC-induced rat model, through the measurements of intracellular ROS, DNA damage, and antioxidant enzyme levels. Rats (*n* = 45) are divided into two groups: control (*n* = 12) and VC-induced (*n* = 33). After two months, VC rats receive lycopene (4 mg/kg and 10 mg/kg) therapy for two months. The results indicate that ROS, DNA damage, superoxide dismutase (SOD), sperm concentration, and motility are significantly increased, compared with the control group. Rats supplemented with 10 mg/kg present increased sperm concentration and catalase (CAT) activity, and a reduction in ROS and DNA damage, compared with VC group. This experimental data confirms that lycopene supplemented therapy protects sperm from OS and sperm DNA damage, by increasing antioxidant activity in rats [104]. Different studies also investigate the beneficial effect of lycopene in infertile men with VC. Gupta and coworkers evaluate the effect of oral lycopene therapy in 30 men with idiopathic, non-obstructive oligoasthenoteratozoospermia, by administering 2000 mcg of lycopene, twice a day for three months. Semen analysis shows an improvement in different semen parameters, such as sperm concentration (20 patients), motility (16 patients), and sperm morphology (14 patients). These results demonstrate a possible role for oral lycopene therapy to increase the semen parameters of infertile men with diagnosed VC [107].

### 3.5. Kallikrein Therapy

Kallikreins, a family of proteases discovered for the first time in 1930 in human pancreatic extracts, are characterized by two different isoforms: a plasma kallikrein, circulating in the blood, and tissue kallikreins, expressed in multiple type of tissues. Dysfunctions in the tissue-specific regulation of kallikreins activity is linked to several pathologies, including pathological inflammation and infertility [108]. Kallikrein is a vasoactive substance, which improves testicular arterial vasodilatation and the function of the sperm membrane, restoring semen parameters in patients with VC [109]. Recent studies highlight the importance of kallikrein therapy for the restoration of male fertility, and report an improvement in intratesticular testosterone and sperm maturation [109]. In a randomized study, Mićić and colleagues identify a sperm motility improvement in infertile men with VC undergoing kallikrein therapy. In this study, a group of 65 oligoasthenozoospermic men are divided in treated (*n* = 38) and placebo (*n* = 27) groups. Patients treated with 600 units of kallikrein for 3 days show an improvement in sperm parameters, such as motility (24% to 35%) and morphology (58% to 71%) [110].

### 3.6. Pentoxifylline Therapy

Pentoxifylline (PTX) is a derivate of methylxanthine, and a non-specific inhibitor of phosphodiesterase. This is considered a potent vasodilator with immunomodulatory effects, leading to reduction in the secretion of inflammatory cytokines such as TNF-α [111]. Recent studies also demonstrate the potent antioxidant activity of PTX, due to its ability to modulate intracellular 3′-5′-cyclic adenosine monophosphate (cAMP) signaling pathways [112]. Indeed, Whang and colleagues, in an experimental study, evaluate its antioxidant activity in a rat model. The results show an increase in the antioxidative capability and mitochondrial biogenesis in rats treated with PTX [113]. In another study, Oliva and colleagues investigate the efficacy of PTX on the semen of 36 men with VC-associated infertility, orally administered with a daily dose of 1200 mg (600 mg every 12 h for 12 days) of PTX. Semen samples are evaluated after 4, 8, and 12 weeks of the first dose, and 4 weeks after the last dose. The results show an improvement in sperm morphology at week 4, which is maintained during and after treatment. This evidence suggests a drug-related positive effect of PTX on the quality of sperm parameters [114].

### 3.7. Cinnoxicam Therapy

Cinnoxicam, a cyclooxygenase inhibitor, is particularly used as anti-inflammatory agent, and is considered a safe therapy for men with oligoasthenospermia associated with VC [115]. To examine this, Cavallini and colleagues investigate the potential role of cinnoxicam on the restoration of semen parameters in patients with grade 3 VC, in a randomized study. Three groups of patients (*n* = 180) are enrolled, based on the therapy received. Group 1 undergo surgery, group 2 receive cinnoxicam, and group 3 receive a placebo. Analysis of the seminal parameters, such as sperm concentration, WHO class A motility, and morphology, are conducted. The results show cinnoxicam significantly increases sperm concentration, motility, and morphology in patients with oligoasthenospermia associated with VC, as demonstrated by group 2, supplemented with cinnoxicam therapy, compared to the placebo and surgery groups [115].

**Table 1 ijms-23-05233-t001:** List of natural non-enzymatic antioxidants and therapeutical drugs for VC treatment.

**Natural, Non-Enzymatic Antioxidants**			
**Antioxidants**	**Natural Source**	**Main Effects**	**References**
Resveratrol	Grapes, mulberry plants	Anti-inflammatory and anti-apoptotic effects; improvement in sperm count, concentration, and motility	[82] [84]
Vitamin E	Vegetable oil, fruits, nuts, seeds	Anti-oxidant effect; increases semen parameters; prevention of DNA damage	[88] [89]
Vitamin C	Fruits, vegetables	Anti-oxidant and anti-inflammatory effects; restoration of sperm motility and morphology	[89] [90]
CoQ10	Oil fish, organ meats, whole grain	Anti-oxidant effect; improvement in sperm motility, ejaculation volume, and sperm density; restoration of mitocondrial function	[94] [95] [96]
Ubiquinol	Oil fish, organ meats, whole grain	Anti-oxidant effect; improvement in sperm motility, ejaculation volume, and sperm density; restoration of mitocondrial function	[97] [98] [99] [100] [101] [102] [103]
Lycopene	Vegetables, fruits	Anti-oxidant effect; increases sperm parameters; protection from DNA damage	[104] [105] [106] [107]
**Therapeutical, non-enzymatic, drugs**			
**Antioxidants**	**Chemical products**	**Main effects**	**References**
Kallikrein	Drug	Arterial vasodilator; improvement in semen parameters, testicular testosterone, and sperm maturation	[108] [109] [110]
Pentoxifylline	Drug	Potent vasodilator; immunodulatory effect; anti-oxidant activity; improvement in sperm morphology	[111] [112] [113] [114]
Cinnoxicam	Drug	Anti-inflammatory effect; improvement in semen parameters	[115]

## 4. Conclusions

The present study described how NLRP3 inflammasome and OS are interlinked in the pathogenesis of VC. The activation of this specific platform leads to the accumulation of cytokines in high amounts in semen, affecting the quality of sperm and male fertility. The imbalance of free radical production and cytokines increase could be restored with an antioxidant therapy, thus, improving spermatogenesis and seminal plasma clearance capacity. However, more studies are needed to understand all of the mechanisms underlying the involvement of NLRP3 inflammasome and OS in the induction of VC, and the consequent VC-induced hypofertility. Moreover, given the lack of literature about the role of VitE, VitC, CoQ10, and lycopene on the modulation of NLRP3 inflammasome in VC, it would be interesting to carry out studies in vivo and in vitro, to characterize their regulatory potential on the NLRP3 inflammasome platform.

## Figures and Tables

**Figure 1 ijms-23-05233-f001:**
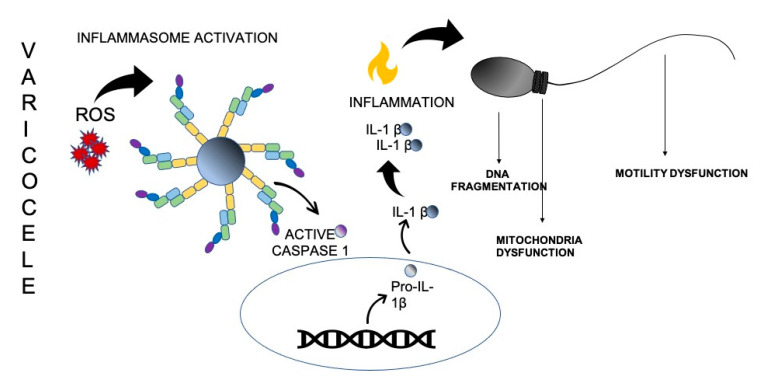
Inflammasome activation and ROS production as core mechanisms linking VC. During the pathogenesis of VC, the testicular hyperthermia, resulting from a dilation and tortuosity of the sperm vein pampiniform plexus, is a source of ROS. This condition activates NLRP3 inflammasome that mediates caspase-1 activation, and the secretion of proinflammatory cytokines, such as IL-1β. ROS generation and inflammasome activation have a negative effect on spermatogenesis, leading to sperm DNA fragmentation, and mitochondrial and motility dysfunction.

## Data Availability

Not applicable.

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
