# Peer review of "The Role of NLRP3 Inflammasome Activation and Oxidative Stress in Varicocele-Mediated Male Hypofertility"

_ijms, 2022, doi:10.3390/ijms23095233_

Round 1

Reviewer 1 Report

General comments

In the review manuscript “The role of NLRP3 inflammasome activation and oxidative stress in varicocele-mediated male hypofertility” the authors describe the role played by oxidative stress, inflammation and NLRP3 inflammasome activation in varicocele disease. They also highlight alternative perspectives by providing ideas for future research and potential treatment of varicocele.

The manuscript is well written. The reader can easily follow every section. The paper results relevant both to the insiders and to biomedical community. The authors have consulted more than 90 publications. I well appreciated the description of possible therapeutic approaches for varicocele targeting oxidative stress and inflammation.

Specific comments

  1. Please, in Introduction decribe OS before illustrating its role in the varicocele-mediated male hypofertility
  2. I recommend to significantly improving the scheme proposed in figure 1. It is the only figure present in the manuscript and describe the subject in its various aspects.

Author Response

We are very grateful to reviewer for valuable suggestions on our review. Based on these comments, we revised the entire manuscript. With this purpose to address satisfactorily the comments of the reviewer, we provided a point-by-point responses to the comments that you will find below.

Reviewer 1: Specific comments

  1. Please, in Introduction describe OS before illustrating its role in the varicocele-mediated male hypofertility
  2. I recommend to significantly improving the scheme proposed in figure 1. It is the only figure present in the manuscript and describe the subject in its various aspects.

Response to Reviewer 1

We thank the reviewer for his comments

  1. In the introduction section, a paragraph described OS according to reviewer suggestions, was added. (page 1-2, from line 42 to line 52). Sentences and references were marked in red color
  2. Caption of figure 1 have been modified according to reviewer suggestions. In the second part of the review, a table (Table 1) summarized all antioxidants molecules, was added. For an easy reading, caption of figure 1 and Table 1 were marked in red color.

Reviewer 2 Report

The review of G.Poli et al is devoted to the well-known medical problem of male hypofertility mediated by varicocele (VC). The review is written to evaluate the role of oxidative stress (OS) and NLRP3 inflammasome activation in VC associated pathologies and to show some directions of VC treatment in order to expand the horizons of practical medicine. The connection of the VC with the development of OS and the NLRP3 activation has been described quite convincingly. At the same time, there is practically no description of the possible mechanisms of influence of NLRP3 activation. And though formally this is not the aim of the article I would recommend the authors to include the references to works describing such mechanisms. A good example is the brilliant review of R.Burke et al published in IJMS recently with a broad picture of cellular signal pathways associated with NLRP3 regulation related to problems with organ-transplantation. A less good impression is made by the second half of the review, which highlights the ways of VC treating by antioxidants. The set of active antioxidants is small may be because only natural antioxidants are considered. But an antioxidant must, after all, be evaluated not by its natural origin, but by its effectiveness. And other cases of treatment with artificial but effective antioxidant forms, for instance mitochondrially targeted antioxidants, might be possibly found in literature. Among all the possible impact of oral ubiquinone treatment is highly questionable due to its instability and ability to interact directly with oxygen to produce ROS while mitochondrially targeted quinone-based antioxidant is in membrane surrounding far more stable and reducible because it is naturally reduced by respiratory chain.

Author Response

We are very grateful to reviewer for valuable suggestions on our review. Based on these comments, we revised the entire manuscript. With this purpose to address satisfactorily the comments of the reviewer, we provided a point-by-point responses to the comments that you will find below.

Reviewer 2: Comments

The review of G. Poli et al is devoted to the well-known medical problem of male hypofertility mediated by varicocele (VC). The review is written to evaluate the role of oxidative stress (OS) and NLRP3 inflammasome activation in VC associated pathologies and to show some directions of VC treatment in order to expand the horizons of practical medicine. The connection of the VC with the development of OS and the NLRP3 activation has been described quite convincingly. At the same time, there is practically no description of the possible mechanisms of influence of NLRP3 activation. And though formally this is not the aim of the article I would recommend the authors to include the references to works describing such mechanisms. A good example is the brilliant review of R. Burke et al published in IJMS recently with a broad picture of cellular signal pathways associated with NLRP3 regulation related to problems with organ-transplantation. A less good impression is made by the second half of the review, which highlights the ways of VC treating by antioxidants. The set of active antioxidants is small may be because only natural antioxidants are considered. But an antioxidant must, after all, be evaluated not by its natural origin, but by its effectiveness. And other cases of treatment with artificial but effective antioxidant forms, for instance mitochondrially targeted antioxidants, might be possibly found in literature. Among all the possible impact of oral ubiquinone treatment is highly questionable due to its instability and ability to interact directly with oxygen to produce ROS while mitochondrially targeted quinone-based antioxidant is in membrane surrounding far more stable and reducible because it is naturally reduced by respiratory chain.

Response to Reviewer 2

We sincerely thank the reviewer for his valuable comments on our review.

We added some concepts regarding NLRP3 regulation, as reviewer suggest, including the review of R. Burke [18] (page 2, from line 65 to line 76). Both sentences and references added, were marked in red color for easy reading.

In the second half of the review, we clarified the role of oral ubiquinone treatment, due to its instability and ability to interact directly with ROS, as the reviewer suggest (page 10, from line 378, to line 396). Sentences and references added, were marked in red color

Three paragraphs, 3.5, 3.6 and 3.7 (pages 11-12, from line 424 to line 469) were added to improving the set of active antioxidants. Sentences and references, were marked in red color.
